# Modeling Mathematical Relationship with Weight Loss and Texture on Table Grapes of ‘Red Globe’ and ‘Wink’ during Cold and Ambient Temperature Storage

**DOI:** 10.3390/foods12132443

**Published:** 2023-06-21

**Authors:** Xiaoyan Cheng, Rongxia Li, Youyi Zhao, Yuhe Bai, Yuanling Wu, Peipei Bao, Zijie Huang, Yang Bi

**Affiliations:** 1College of Science, Gansu Agricultural University, Lanzhou 730070, China; 2School of Mathematics and Statistics, Lanzhou University, Lanzhou 730070, China; 3School of Mathematical Sciences, University of Electronic Science and Technology of China, Chengdu 611731, China; 4College of Food Science and Engineering, Gansu Agricultural University, Lanzhou 730070, China

**Keywords:** *Vitis vinifera* L., storage, weight loss, texture, modeling

## Abstract

Weight loss associated with fruit texture during storage has received numerous reports; however, no research has been conducted on the mathematical relationships between weight loss and textural traits of table grapes stored at cold and ambient temperatures. In this study, it was found that the weight loss of ‘Red Globe’ was in the range of 0 to 0.0487, 0 to 0.0284 and 0 to 0.0199 compared to 0 to 0.0661, 0 to 0.0301 and 0 to 0.028 of ‘Wink’ at 13 °C, 3 °C, and 0 °C of storage for 13 days. Stored for 13 days at 13 °C, 3 °C, and 0 °C, the range of the textural traits of failure force, strain and penetration work in ‘Red Globe’ were 6.274 to 3.765, 6.441 to 3.867, 6.321 to 4.014; 51.931 to 11.114, 51.876 to 13.002, 51.576 to 20.892; 21.524 to 13.225, 21.432 to 14.234, 21.321 to 15.198 in contrast to in ‘Wink’ of 4.4202 to 2.2292, 4.4197 to 2.653, 4.4371 to 2.8199 and 15.674 to 2.7881, 15.776 to 4.1431, 15.704 to 5.702 and 12.922 to 7.754, 12.909 to 8.021, 12.915 to 8.407. Meanwhile, the weight loss and textural traits of two table grapes were examined using time-dependent and weight loss-dependent modeling at 13 °C, 3 °C, and 0 °C of storage. The Logistic, ExpDec1, and ExpDec2 models, as well as the Boltzmann model, were identified as the best fit for the obtained data. The equations proved to be more effective in characterizing the change in weight loss and texture of ‘Red Globe’ and ‘Wink,’ with the best equations suited to the weight loss and textural parameters having an average mean standard error of 2.89%. The viability of the established models was evaluated, and parametric confidence intervals of the equations were proposed to fit different grape cultivars. According to the findings, the weight loss and texture of the two grape cultivars could be accurately predicted by the established models; additionally, the results showed that cold storage is better for the quality of table grapes and that weight loss can predict the textural quality of table grapes. This study provides a theoretical framework for optimum storage temperature together with a significantly convenient and quick approach to measure the texture of grapes for fruit dealers and enterprises.

## 1. Introduction

*Vitis vinifera* L., commonly known as table grapes, is widely planted across China and the globe. This fruit is popular and consumed for its fresh and crisp taste as well as its high nutritional value, which is due to its high soluble solids content [1]. However, freshly harvested table grapes are susceptible to quality degradation and short shelf life because of rapid physiological changes and spoilage during postharvest storage, resulting in lower commercial value. It was reported that the weight loss and texture affected the quality of the fruit [2,3]. Meanwhile, weight loss and texture decline play a key role in the storability of fresh produce [4]. Temperature is one of the most important factors affecting the weight loss of grapes during storage [5]. Table grapes showed significant weight loss when stored at room temperature [6]. A high weight loss (over 4.5%) of ‘White Seedless’ was reported during 7 days of storage at 15 °C [7]. In another study, it was found that ‘Crimson Seedless’ experienced a weight loss of 7.63% after 10 days while under room storage (25.1 ± 1.3 °C) [8]. Similarly, ‘Yaghouti’ was found to have a weight loss of 10.43% during 40 days of storage at room temperature [6]. In contrast, the ‘Rishbaba’ grape underwent a lower weight loss of 3% while storing at 0 ± 1 °C with 85% relative humidity for 15 days [1]. At similar relative humidity, fruit recorded less weight loss at low temperatures [9]. Currently, weight loss in table grapes is managed through methods such as chemical treatments and plastic packaging at room and cold temperature storage. However, due to food safety concerns and environmental impacts, there is a need for a better solution that involves determining an appropriate storage time to control weight loss. Recently, mathematical models have been developed in conjunction with computational methods to detect fruit weight loss, which is a highly efficient approach that can potentially increase sales [10,11,12]. The texture is one of the most important factors affecting consumer acceptance of fruit [13]. The texture of table grapes deteriorates rapidly during storage [14]. The hardness of ‘Autumn Rogal’ decreased significantly by 26% during 35 days of storage at 2 °C, resulting in softening and worse taste [15]. After 56 days of storage at 1 °C, the hardness of ‘Aledo’ decreased by nearly 80% [6]. When stored at a cool temperature(1 ± 0.5 °C), the hardness, chewiness, and resilience of ‘Red Globe’ decreased rapidly [16]. The texture loss will shorten the table grapes’ postharvest life causing serious economic loss [17]. However, the detection of texture is complicated and time-consuming. TPA is the main method to detect the texture properties in the lab [18,19], which is not easily accomplished outside of the lab. So, modeling methods were used to describe the textural changes in fruit [20,21]. Research indicates that there is a strong correlation between the texture of fruit and its weight loss. As weight loss decreases, the texture of berries undergoes significant changes. In particular, a decrease in weight loss leads to a rapid increase in the hardness, resilience, and chewiness of the fruit [22]. The chewiness of ‘Centurion’ blueberries declined while the weight loss increased [23]. Weight loss of cherry tomatoes ‘Trebus’ and ‘Dorotea’ was strongly associated with their hardness, which decreased seriously with the increased weight loss [24]. Weight loss has a notable impact on the brittleness, springiness, and cohesiveness of fresh Goji berries. As weight loss increases, there is a gradual reduction in the brittleness, springiness, and cohesiveness of the fruit [25]. Weight loss drastically reduced the hardness, cohesiveness, chewiness, and resilience of MAP-packed cherry tomatoes [26]. Mathematical models can be used to deduce quantitative function relationships. Valderrama et al. [27] used a semi-quantitative model to evaluate the presence of carbendazim in grape juices. The model’s robustness is satisfactory, and the outcome would contribute to improving quality control in the industry. The grape photosynthetic data were validly conducted by the Freundlich model, which provides theoretical confirmation that the grape leaves adapt to the environment better under a single oblique cordon [28]. Mathematical models have also been useful in predicting the weight loss and texture of MAP fruits based on storage time and weight loss [26].

Although weight loss and texture changes during storage and their relationship with berries have been reported, there are few reports on mathematical relationships between weight loss and texture characteristics of table grapes, and there is no convenient and fast method to estimate the texture of grapes. In this study, we aimed to (1) investigate the weight loss and texture characteristics of ‘Red Globe’ and ‘Wink’ during ambient and cold temperature storage; (2) analyze the correlation between weight loss and textural characteristics; (3) establish mathematical models for weight loss and textural parameters; (4) verify the reliability of the established models and find a simple way to determine texture quality.

## 2. Materials and Methods

### 2.1. Grapes

Fresh table grapes (*Vitis vinifera* L. cvs. Red Globe and Wink) were obtained at the maturity stage (°brix: Red Globe, 17–18%, Wink, 15–16%; total acid: Red Globe, 6.206 g/L–6.296 g/L, Wink, 5.769 g/L–6.522 g/L; sugar-acid ratio: Red Globe, 27–29, Wink, 23–26; firmness: Red Globe, 128.299 g mm, Wink, 161.049 g mm) from the local market (altitude, 1531 m; longitude, 103.706090; latitude, 36.099183) in Lanzhou, China. The fruit was selected based on the following features; same size with a round shape, consistency of bright color, devoid of mechanical damage, presence of green stems, and moderate acidity and sweetness. Approximately 3 kg of grapes were packed per bag using polyethylene (PE) bags (Anhui Tongcheng Xiangpeng Packaging Co., Ltd., Tongcheng, China) (thickness, 0.018 mm, 32 cm × 54 cm) and stored at 13 °C, 3 °C, and 0 °C (RH 85~90%) for use.

### 2.2. Weight Loss

Weight loss was determined by the gravimetric method (TC30K-H, Jinboshi (Suzhou, China, Biotechnology Co., Ltd., Beijing, China). The weight of grapes was measured at 0, 1, 3, 5, 7, 9, 11, and 13 days during storage, and the weight loss is expressed according to the formula: weight loss = [(pre-storage weight − post-storage weight)/pre-storage weight] [29]. Each experiment was repeated three times at least.

### 2.3. Texture

Texture Profile Analysis test was based on a previous study [30]. Texture Analyser (TA.XT Express, SMS (Stable Micro Systems), Godalming, UK) with a P/2 probe the 2 mm diameter was used to measure the textural parameters following the operating setting: pre-test speed, 1.00 mm s^−1^, test speed, 1.00 mm s^−1^, post-test speed, 6.00 mm s^−1^, compression distance10.0 mm, 5.0 s rest period between cycles, and 5.0 g target force. Each experiment was carried out on 13 samples in triplicates at least.

### 2.4. Statistical Analysis

Weight loss and texture results were analyzed using Origin 2021 (Origin Lab Corporation, Northampton, MA, USA) and presented as the means ± SD (standard deviations). Pearson’s correlation coefficient and Duncan’s multiple comparison tests were performed using SAS 9.4 (SAS Institute Inc., Cary, NC, USA). A significant correlation was assumed for *p* < 0.05.

### 2.5. Characteristics of Weight Loss Based on the Logistic Model

A Logistic model is a nonlinear model, which can describe the decreasing or increasing trend of dependent variable with the independent variable. The Logistic model is expressed as follows [31]
(1)$$y=y_{0}+\frac{a_{0}-y_{0}}{1+{\left(\frac{x}{x_{0}}\right)}^{p}}$$
where y is the weight loss for ‘Red Globe’ and ‘Wink’; x is the storage time, $y_{0}$ is the baseline, $a_{0},\;x_{0}$ are the constants of the equations with the storage time.

### 2.6. Characteristics of Textural Parameters Based on Logistic Model, ExpDec1 Model, ExpDec2 Model, and Boltzmann Model

Accurate prediction of fruit texture is significant in freshness. The Logistic equation, ExpDec1 equation, ExpDec2 equation and Boltzmann equation were expressed here for describing the changes in texture [32]. The textural parameters (failure force, strain, penetration work) under storage time and weight loss can be presented as follows: Equation (1),
(2)$$y=A_{1}\cdot e^{-\frac{x}{t_{1}}}+y_{0}$$
(3)$$y=A_{1}\cdot e^{-\frac{x}{t_{1}}}+A_{2}\cdot e^{-\frac{x}{t_{2}}}+y_{0}\;$$
(4)$$y=A_{2}+\frac{A_{1}-A_{2}}{1+e^{\frac{x-x_{0}}{dx}}}\;$$

Equation (1) developed for calculating the failure force, strain, and penetration work with storage time under 13 °C, 3 °C, and 0 °C for ‘Red Globe’ and ‘Wink’, which is also used for calculating the failure force and strain with weight loss for ‘Wink’ at 3 °C and for ‘Red Globe’ at 3 °C and 0 °C. The penetration work with weight loss for ‘Wink’ at 0 °C and ‘Red Globe’ at 13 °C, 3 °C, and 0 °C was calculated by Equation (1) too. Equation (2) was developed for calculating the strain and penetration work for ‘Wink’ at 13 °C. Equation (3) was developed for calculating the failure force with weight loss for ‘Wink’ at 13 °C. Equation (4) was developed for calculating the failure force, and strain with weight loss for ‘Wink’ at 0 °C and for ‘Red Globe’ at 13 °C, and it is also used to calculate the penetration work with weight loss for ‘Wink’ at 3 °C.

## 3. Results and Discussion

### 3.1. Changes in Weight Loss in Two Cultivars during Cold and Ambient Storage

Weight loss is a key index for evaluating the postharvest quality of table grapes [33]. The weight loss of the two cultivars increased gradually during the storage period. Both ‘Wink’ and ‘Red Globe’ kept lower weight loss under cold storage. During the first 5 days of storage, there was practically no difference in weight loss between the two cultivars at 0 °C, at the late storage, the weight loss of ‘Red Globe’ was 29% lower than ‘Wink’. At 3 °C, the weight loss of ‘Red Globe’ is essentially the same with ‘Wink’ storing for 7 days; however, the weight loss of ‘Wink’ is 5.5% higher than ‘Red Globe’ during the late period of storage. By contrast, the ambient temperature caused the higher weight loss of two cultivars’ grapes. During the 13 °C temperature storage, the weight loss of ‘Red Globe’ was a little higher than ‘Wink’ after the first 3 days, which may be caused by the storage environment, however, the weight loss of ‘Red Globe’ was significantly lower than ‘Wink’ during the middle and late storage stage (Figure 1A). Low temperature delays the weight loss process of grapes because low temperature reduces the postharvest respiratory intensity and inhibits the activities of various enzymes in the body [34]. The Logistic models in Origin 2021 were used to fit the change in weight loss for ‘Wink’ and ‘Red Globe’ (Figure 1B). The equations were shown in Table 1, and R^2^ correlation coefficients were all greater than 0.90, which means the fitting is significant. The confidence interval of fitting coefficients is shown in Table 2. Through the established equations, the prediction values of the weight loss for ‘Wink’ and ‘Red Globe’, can be calculated. The confidence interval of Logistic fitting coefficients of weight loss can be used to adjust the prediction accuracy on ‘Wink’ and ‘Red Globe’ (Table 2). It was found that the deviations between the measured values and the forecasting values of the models were less than 2.56%. Weight loss in fresh fruit mainly occurs through the reduction in moisture loss due to respiration and transpiration [35,36]. The moisture content of ‘Wink’ was much higher than the ‘Red Globe’ in total weight [37], resulting in wilting of fruit peel, tissue disintegration, and loss of freshness [38]. The weight loss of ‘Wink’ was higher than ‘Red Globe’, which was likely due to the surface area and thickness of the berry peel [39]. In terms of weight loss, ‘Red Globe’ showed a better shelf life than ‘Wink’.

### 3.2. Changes of Textural Parameters in Two Cultivars during Cold and Ambient Storage

Textural properties reflect the physical properties of fruit as they relate to mechanical properties. To some extent, the measured indices can represent textural qualities and structural changes, and they can be used to evaluate the textural quality of fruit stored at different temperatures (cold and ambient). Texture is an important quality element that influences the consumption of table grapes [40]. Failure force, strain, and penetration work are major textural indicators [41,42]. The failure force of fruit involving tissue and maturity is the maximum stress sustained by fruit as measured by the probe [43]. During the storage time, the failure force of ‘Red Globe’ was better than ‘Wink’, which showed decreases of 39.73% and 49.57%, 37.46% and 42.65%, 36.49% and 36.44% for ‘Red Globe’ and ‘Wink’ at the temperature 13 °C, 3 °C, 0 °C. So, the failure force of ‘Red Globe’ maintained a higher level than that of ‘Wink’ during storage (Figure 2A). Strain refers to the local relative deformation of an object under the action of external force and a non-uniform temperature field [44,45]. The strain of ‘Red Globe’ and ‘Wink’ decreased during the days of storage; however, the ‘Red Globe’ maintained a relatively slow decrease with 78.59%, 74.94%, and 59.49% compared with 82.21%, 75.74%, 63.69% in ‘Wink’ at 13 °C, 3 °C, and 0 °C from the first day to the end of storage (Figure 2C). The trend of strain was fitted by Logistic models in both ‘Red Globe’ and ‘Wink’ (Figure 2D). Penetration work was carried out by the penetration force, which is displaced in the direction of the probe force [46]. Textural analysis showed that penetration work decreased rapidly during the storage in both ‘Red Globe’ and ‘Wink’ (Figure 2E); however, the amount of variability is different, the decrement of ‘Red Globe’ was 38.56%, 33.59%, 28.72% in comparison with 39.99%, 37.87%, 34.91% of ‘Wink’ at the 13 °C, 3 °C and 0 °C. The reduction in failure force, strain, and penetration work of ‘Red Globe’ and ‘Wink’ was probably due to the decline in moisture and pectin levels [47]. Pectin exists in fruit tissues in three different forms: protopectin, pectin, and pectin acid. As the fruit ripens, the protopectin is broken down into pectin by the protopectinase, which dissolves in water and makes the texture of the fruit worsen [48]. With the increased storage time, the content of total sugar and the total acid decreased gradually [49], which was obvious on ‘Wink’. This may be the reason for the higher decline in texture on ‘Wink’. Meanwhile, it was found that the change in failure force, strain, and penetration work best fit the Logistic models during the storage time 13 °C, 3 °C, 0 °C (Figure 2B,D,F), the fitting models and the goodness of fit are shown in Table 3, from which it was found that all the values of goodness of fit are greater than 0.98, indicating the validity of the established models. The estimated set of totality parameters established by sampling statistics is referred to as the confidence interval. The confidence interval for a probability sample is an interval estimate of a totality parameter for the sample in statistics. The confidence interval shows the degree to which the actual value of the parameter falls with a specific probability (*p* < 0.05) around the measured result. Grape fruit failure force, strain, and penetration work can be predicted without experiment using existing models and confidence intervals of model parameters (Table 4). It was also found that the textural change in ‘Red Globe’ was more stable than ‘Wink’ during ambient storage, which may be attributed to the size of intercellular adhesion, mechanical strength of the cell wall, and cell expansion [41,42]. The results indicated that ‘Red Globe’ is more suitable for long storage and that cold temperatures are beneficial for table grape storage, confirming similar findings by Nunes et al. [50].

### 3.3. Correlation between Weight Loss and Textural Parameters of Two Cultivars during Ambient and Cold Storage

Correlation analysis examines the interdependence of two variables. The correlation coefficient is a metric that measures the degree and direction of a linear relationship. The direction of correlation is represented by positive and negative symbols, while the strength of the correlation is represented by the absolute value. Figure 3 illustrates the correlation between weight loss and failure force, strain, and penetration work of ‘Red Globe’ and ‘Wink’ at 13 °C, 3 °C, and 0 °C storage, respectively. The weight loss of ‘Red Globe’ and ‘Wink’ was negatively related to failure force, strain, and penetration work while the *p* < 0.05 with the minimum absolute value of 0.81. Failure force, strain, and penetration work were all positively correlated with the least value of 0.92 under cold and ambient temperatures. The results showed that the weight loss had negative influence on texture, and the textural parameters had the same trend of change with each other. Similar results were reported on cherry tomatoes [26].

### 3.4. Changes and Fitting Models of Textural Parameters with Weight Loss of Two Cultivars during Ambient and Cold Storage

With the increase in weight loss, failure force, strain and penetration work declined gradually. At 13 °C, 3 °C, and 0 °C storage, when the weight loss was within the range of 0 to 0.04765, 0 to 0.0297, and 0 to 0.0198, the loss of failure force, strain, and penetration work in ‘Red Globe’ was 38.75%, 34.50%, 30.49%; 78.59%, 63.94%, 50.25%; 38.56%, 33.59%, 28.72% compared to 49.04%, 35.01%, 32.33%; 79.34%; 65.03%, 56.06%; 37.69%, 34.68%, 30.99% in ‘Wink’ (Figure 4A,C,E). A mathematical model is a kind of mathematical structure that is expressed in general or approximate terms by referring to the characteristics or quantitative dependence of a particular system using mathematical language. This kind of mathematical structure is a pure relational structure of a certain system drawn with the help of mathematical symbols. To establish the fitting models, the dependent variables were failure force, strain, and penetration work, and the independent variable was weight loss. The Logistic model, ExpDec2 model, ExpDec1 model and Boltzmann model in Origin 2021 were used to fit failure force, strain, and penetration work with the weight loss in ‘Red Globe’ and ‘Wink’ (Figure 4B,D,F). The fitting equations were shown in Table 5 and all the correlation coefficients were greater than 0.89, which showed the goodness of the fitting effect. Through models and confidence intervals of parameters (Table 6), predicted values of the failure force, strain, and penetration work can be calculated and it was found that the prediction was very close to those measured. Through the fitting models based on weight loss, the change in textural indicators could be evaluated, and all the above results reflect the importance of weight loss on texture. The difference in textural expression between ‘Red Globe’ and ‘Wink’ was probably caused by the different moisture and pectin content of the two cultivars [51,52]. Thus, we conclude that weight loss determines the textural quality of table grapes.

## 4. Conclusions

Weight loss of ‘Red Globe’ and ‘Wink’ increased, and the textural quality deteriorated during ambient and cold storage. However, the weight loss and textural quality of ‘Red Globe’ were much lower and better than those of ‘Wink’, indicating that the ‘Red Globe’ is more suitable for storage, while the cold temperature is also beneficial. The weight loss of ‘Red Globe’ and ‘Wink’ increased according to Logistic models and negatively correlated with failure force, strain, and penetration work significantly. The failure force, strain, and penetration work of ‘Red Globe’ and ‘Wink’ decreased with increasing storage time via Logistic models. Selected textural parameters (failure force, strain, and penetration work) were chosen as dependent variables, and Logistic mathematical models, ExpDec2 models, ExpDec1 models, and Boltzmann models were developed based on weight loss. The accuracy of fit in these models was much higher. The change in weight loss and textural traits in ‘Red Globe’ was lower than those in ‘Wink’. The established models can accurately determine the weight loss and textural parameters of grapes. This study can predict changes in weight and textural quality of table grapes stored at ambient and cold temperatures. Moreover, the research suggests that the textural quality of the grapes can be speculated by weight loss, which could provide a theoretical foundation for developing new rapid and convenient textural detection equipment. The outcome may be applicable to other berries as well, which should be confirmed in future research.

## Figures and Tables

**Figure 1 foods-12-02443-f001:**
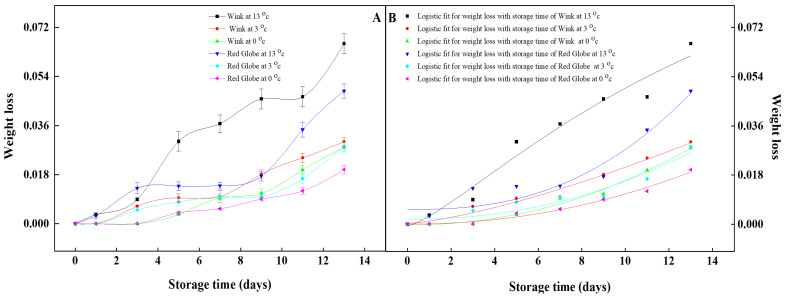
Variations of weight loss in ‘Wink’ and ‘Red Globe’ during storage (**A**) and their fitting curves against storage time (**B**). Vertical bars represent standard deviation of the mean (±SD).

**Figure 2 foods-12-02443-f002:**
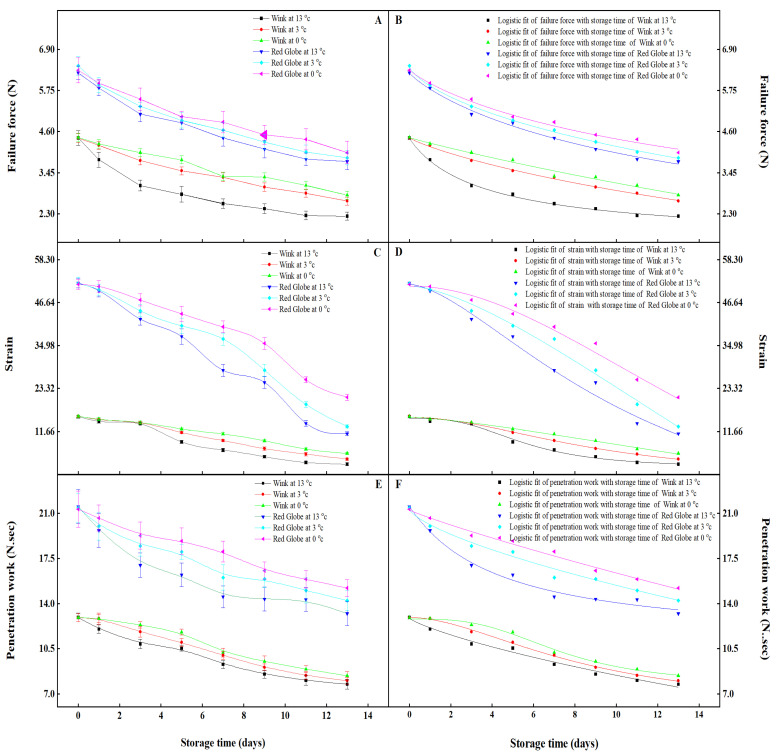
Variations of failure force (**A**), strain (**C**), penetration work (**E**), fitting of failure force of storage time (**B**), fitting of failure strain of storage time (**D**), fitting of penetration work of storage time (**F**) in ‘Red Globe’ and ‘Wink’ at 13 °C, 3 °C and 0 °C storage. Vertical bars represent standard deviation of the mean (±SD).

**Figure 3 foods-12-02443-f003:**
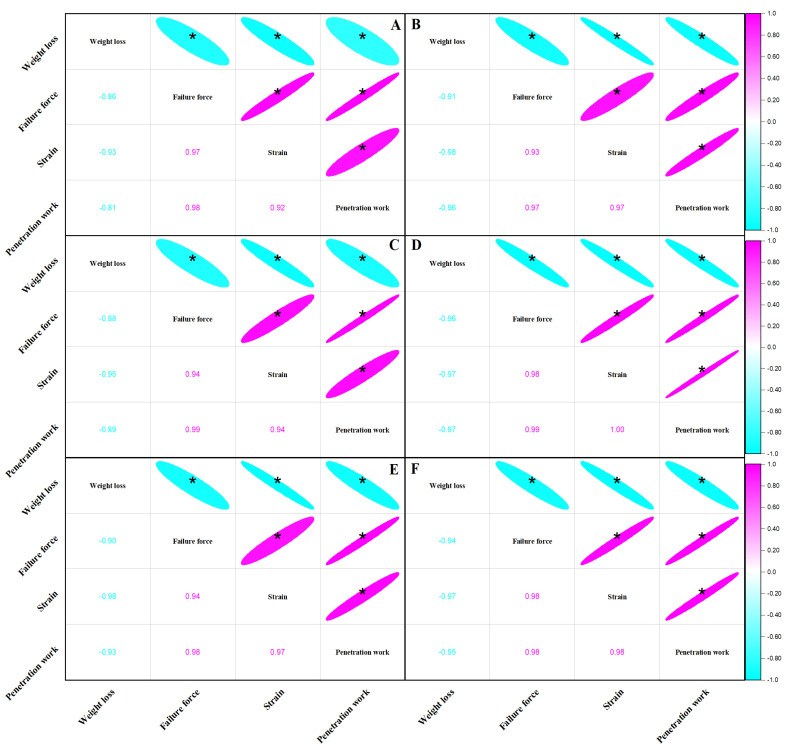
Correlation between weight loss and failure force, strain, and penetration work of ‘Red Globe’ and ‘Wink’ at 13 °C (**A**,**B**), 3 °C (**C**,**D**), and 0 °C (**E**,**F**) storage. Note: * indicates a significant correlation (*p* < 0.05).

**Figure 4 foods-12-02443-f004:**
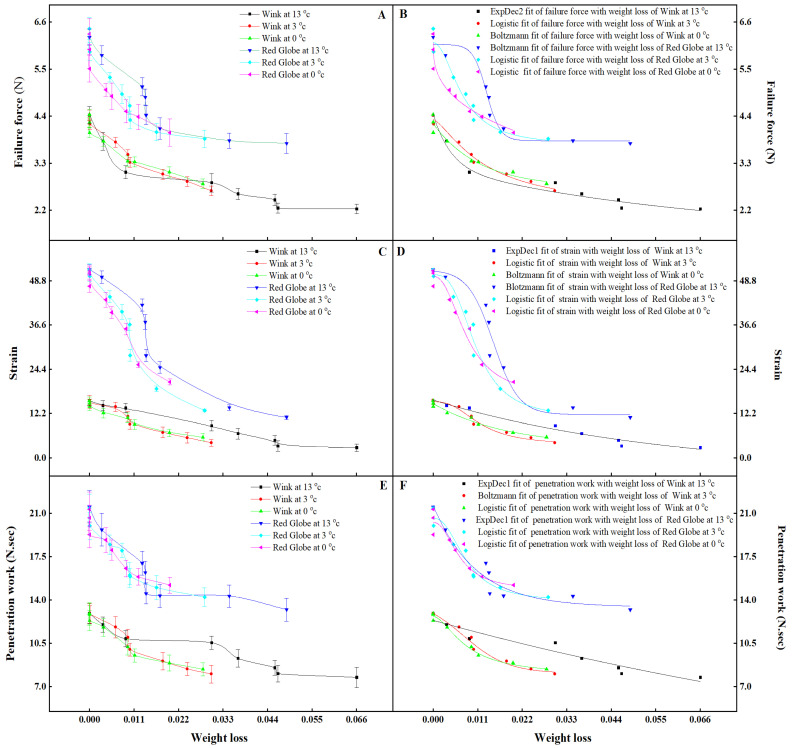
Variations of failure force (**A**), strain (**C**), penetration work (**E**), fitting of failure force with weight loss (**B**), fitting of failure strain with weight loss (**D**), fitting of penetration work with weight loss (**F**) in ‘Red Globe’ and ‘Wink’ at 13 °C, 3 °C and 0 °C storage. Vertical bars represent standard deviation of the mean (±SD).

**Table 1 foods-12-02443-t001:** The fitting equations of weight loss with storage time.

Cultivar	Temperature	Equation	R-Square	Adj. R-Square
	13 °C	$y=341.15583-\frac{341.15045}{1+{\left(\frac{x}{459.51569}\right)}^{2.52063}}$	0.93158	0.88026
Red Globe	3 °C	$y=364.65156-\frac{364.64973}{1+{\left(\frac{x}{1109.44697}\right)}^{2.16026}}$	0.93102	0.87928
	0 °C	$y=88.86516-\frac{88.86514}{1+{\left(\frac{x}{760.47948}\right)}^{2.07585}}$	0.98425	0.97243
Wink	13 °C	$y=0.14365-\frac{0.14427}{1+{\left(\frac{x}{16.03276}\right)}^{1.31641}}$	0.96782	0.94367
	3 °C	$y=560.90825-\frac{560.90793}{1+{\left(\frac{x}{15988.71322}\right)}^{1.38471}}$	0.98071	0.96625
	0 °C	$y=29.46132-\frac{29.46158}{1+{\left(\frac{x}{360.23485}\right)}^{2.09308}}$	0.98973	0.98202

**Table 2 foods-12-02443-t002:** The confidence interval of model fitting coefficients of weight loss.

Cultivars	Coefficient	Temperature		
		13 °C	3 °C	0 °C
Red Globe	$a_{1}$	[0.00464,0.00612]	[0.00152,0.00214]	$\left[-2.96\times 10^{-5},7.121\times 10^{-5}\right]$
	$a_{2}$	[333.74498,348.56668]	[354.87128,374.43184]	[85.28762,92.44270]
	$x_{0}$	[453.54337,465.48801]	[1108.122,1110.768]	[746.99875,773.96021]
	p	[2.46077,2.58049]	[1.69087,2.62965]	[1.97038,2.18132]
Wink	$a_{1}$	$\left[5.43\times 10^{-4},-7.03\times 10^{-4}\right]$	$\left[3.14\times 10^{-4},3.18\times 10^{-4}\right]$	$\left[-2.63\times 10^{-4},-2.61\times 10^{-4}\right]$
	$a_{2}$	[0.12447,0.16283]	[554.98,566.83]	[25.36,33.56]
	$x_{0}$	[12.75,19.31]	[15986.74,15990.69]	[348.68,371.78]
	p	[1.27061,1.36221]	[1.36381,1.40561]	[1.99621,2.18995]

**Table 3 foods-12-02443-t003:** The fitting equations of textural parameters for ‘Red Globe’ and ‘Wink’ with storage time.

Cultivar	Parameter	Temperature	Equation	R-Square	Adj. R-Square
Red Globe	Failure Force	13 °C	$y=0.52224+\frac{5.72763}{1+{\left(\frac{x}{16.58117}\right)}^{0.8694}}$	0.99482	0.99094
		3 °C	$y=-4.43593+\frac{10.88153}{1+{\left(\frac{x}{71.48013}\right)}^{0.68091}}$	0.99892	0.99812
		0 °C	$y=-5.55026+\frac{11.87674}{1+{\left(\frac{x}{87.93426}\right)}^{0.75878}}$	0.99417	0.98979
	Strain	13 °C	$y=-21.56186+\frac{73.13646}{1+{\left(\frac{x}{11.22858}\right)}^{1.5896}}$	0.99218	0.98632
		3 °C	$y=-125.3548+\frac{176.48044}{1+{\left(\frac{x}{29.87047}\right)}^{1.54407}}$	0.99591	0.99284
		0 °C	$y=-37.65639+\frac{88.79051}{1+{\left(\frac{x}{18.02382}\right)}^{1.95951}}$	0.99234	0.98659
	Penetration work	13 °C	$y=10.92112+\frac{10.62339}{1+{\left(\frac{x}{4.27226}\right)}^{1.01703}}$	0.98164	0.96787
		3 °C	$y=-2895.20513+\frac{2916.55608}{1+{\left(\frac{x}{86083.88841}\right)}^{0.68371}}$	0.98102	0.96679
		0 °C	$y=-100.19625+\frac{121.45384}{1+{\left(\frac{x}{322.47497}\right)}^{0.9131}}$	0.98823	0.9794
Wink	Failure Force	13 °C	$y=1.50449+\frac{2.92088}{1+{\left(\frac{x}{3.845}\right)}^{0.93451}}$	0.99700	0.99476
		3 °C	$y=-2.59917+\frac{7.02921}{1+{\left(\frac{x}{46.81189}\right)}^{0.8565}}$	0.99779	0.99613
		0 °C	$y=-6.51257+\frac{10.9439}{1+{\left(\frac{x}{81.41922}\right)}^{0.96749}}$	0.98567	0.97493
	Strain	13 °C	$y=1.87317+\frac{13.31486}{1+{\left(\frac{x}{5.5739}\right)}^{2.95782}}$	0.99243	0.98675
		3 °C	$y=-1.59543+\frac{17.07508}{1+{\left(\frac{x}{9.19699}\right)}^{1.9645}}$	0.99710	0.99492
		0 °C	$y=-37.20831+\frac{52.75268}{1+{\left(\frac{x}{37.06981}\right)}^{1.38597}}$	0.99608	0.99413
	Penetration work	13 °C	$y=-29.76248+\frac{42.63276}{1+{\left(\frac{x}{142.86896}\right)}^{0.8108}}$	0.98275	0.96981
		3 °C	$y=5.70461+\frac{7.22033}{1+{\left(\frac{x}{8.47879}\right)}^{1.80672}}$	0.99866	0.99765
		0 °C	$y=7.74001+\frac{5.06346}{1+{\left(\frac{x}{7.31046}\right)}^{3.12745}}$	0.99281	0.98742

**Table 4 foods-12-02443-t004:** The confidence interval of coefficients on textural parameters for ‘Red Globe’ and ‘Wink’ with storage time.

Cultivar	Parameter	Coefficient	Temperature		
			13 °C	3 °C	0 °C
Red Globe	Failure force	$a_{1}$	[6.16993,6.32981]	[6.39828,6.49292]	[6.24613,6.40683]
		$a_{2}$	[0.46624,0.57824]	[−11.57484,2.70298]	[−6.38502,−4.71550]
		$x_{0}$	[16.41186,16.75048]	[69.75094,73.20932]	[83.96028,91.90824]
		p	[0.63947,1.09933]	[0.59014,0.77168]	[0.52668,0.99088]
	Strain	$a_{1}$	[49.48241,53.66679]	[49.47403,52.77725]	[49.82682,52.44142]
		$a_{2}$	[−25.29523,−17.82849]	[−128.46816,−122.24144]	[−38.98675,−36.60300]
		$x_{0}$	[10.05313,12.40403]	[24.80051,34.94043]	[16.10789,19.93975]
		p	[1.5381,1.6411]	[1.486823,1.601317]	[1.88498,2.03404]
	Penetration work	$a_{1}$	[20.95924,22.12978]	[20.80332,21.89858]	[20.79289,21.72229]
		$a_{2}$	[10.12217,11.72007]	[−2911.51023,−2878.90003]	[−111.36209,−89.03041]
		$x_{0}$	[3.87114,4.67338]	[−44596916,44796.083]	[271.3253,373.62464]
		p	[0.93540,1.09866]	[0.65458,0.71284]	[0.4275,1.3987]
Wink	Failure force	$a_{1}$	[4.35925,4.49149]	[4.38951,4.47057]	[4.33912,4.52354]
		$a_{2}$	[1.18454,1.82444]	[−8.60697,3.40863]	[−7.13616,−5.88898]
		$x_{0}$	[2.84769,4.84231]	[45.09059,48.53319]	[76.74624,86.09220]
		p	[0.88217,0.98685]	[0.79934,0.91336]	[0.89008,1.04490]
	Strain	$a_{1}$	[14.70333,15.67273]	[15.24604,15.71326]	[15.23907,15.84967]
		$a_{2}$	[0.79486,2.95148]	[1.609951,−1.580909]	[−38.43349,−35.98313]
		$x_{0}$	[5.06594,6.08186]	[8.34757,10.04641]	[31.48554,42.65408]
		p	[2.34502,3.57062]	[1.93997,1.98903]	[1.34963,1.42231]
	Penetration work	$a_{1}$	[12.54387,13.19669]	[12.84198,13.00790]	[12.64815,12.95879]
		$a_{2}$	[−32.75160,26.77336]	[5.67263,5.73659]	[7.08097,8.39905]
		$x_{0}$	[130.14551,155.59241]	[8.33195,8.62563]	[6.51088,8.11004]
		p	[0.76868,0.85292]	[1.78748,1.82596]	[2.43083,3.82407]

**Table 5 foods-12-02443-t005:** The fitting equations of textural parameters for ‘Red Globe’ and ‘Wink’ with weight loss.

Cultivar	Parameter	Temperature	Equation	R-Square	Adj. R-Square
Red Globe	Failure force	13 °C	$y=3.82412+\frac{2.25238}{{1+e}^{\left(\frac{x-0.0132}{0.00167}\right)}}$	0.97083	0.94895
		3 °C	$y=3.69409+\frac{2.42122}{1+{\left(\frac{x}{0.00718}\right)}^{2.02752}}$	0.96500	0.93875
		0 °C	$y=-610.78964+\frac{616.89872}{1+{\left(\frac{x}{136452.13277}\right)}^{0.36248}}$	0.97180	0.95065
	Strain	13 °C	$y=11.87432+\frac{39.97961}{1{+e}^{\left(\frac{x-0.01485}{0.00314}\right)}}$	0.97837	0.96215
		3 °C	$y=10.91451+\frac{39.5606}{1+{\left(\frac{x}{0.01054}\right)}^{2.93094}}$	0.98895	0.98067
		0 °C	$y=16.55786+\frac{33.48001}{1+{\left(\frac{x}{0.00835}\right)}^{2.28956}}$	0.98358	0.97126
	Penetration work	13 °C	$y=8.19668\times e^{-\frac{x}{0.01056}}+13.43329$	0.89864	0.8581
		3 °C	$y=13.83602+\frac{6.75875}{1+{\left(\frac{x}{0.00818}\right)}^{2.37295}}$	0.93022	0.87788
		0 °C	$y=14.78461+\frac{5.47665}{1+{\left(\frac{x}{0.00648}\right)}^{2.23696}}$	0.96170	0.93298
Wink	Failure force	13 °C	$y=1.22787\times e^{-\frac{x}{0.00426}}+1.61412\times e^{-\frac{x}{0.06549}}+1.60645$	0.94455	0.93703
		3 °C	$y=2.21049+\frac{2.11022}{1+{\left(\frac{x}{0.01271}\right)}^{1.39295}}$	0.98469	0.9732
		0 °C	$y=2.74437+\frac{817.02009}{{1+e}^{\left(\frac{x-0.07239}{0.01147}\right)}}$	0.95667	0.92417
	Strain	13 °C	$y=19.93017\times e^{-\frac{x}{0.05813}}-4.03177$	0.97816	0.96943
		3 °C	$y=3.51968+\frac{11.91177}{1+{\left(\frac{x}{0.01151}\right)}^{2.59173}}$	0.97830	0.96021
		0 °C	$y=3.19295+\frac{299.26602}{1{+e}^{\left(\frac{x+0.05606}{0.01751}\right)}}$	0.97821	0.96186
	Penetration work	13 °C	$y=14.35885\times e^{-\frac{x}{0.15673}}-1.99883$	0.90820	0.87148
		3 °C	$y=8.08817+\frac{6.07897}{1{+e}^{\left(\frac{x-0.00747}{0.00554}\right)}}$	0.97018	0.96723
		0 °C	$y=7.93493+\frac{4.7434}{1+{\left(\frac{x}{0.00818}\right)}^{1.72073}}$	0.98940	0.98144

**Table 6 foods-12-02443-t006:** The confidence interval of coefficients on textural parameters for ‘Red Globe’ and ‘Wink’ with weight loss.

Cultivar	Parameter	Coefficient			Temperature		
					13 °C	3 °C	0 °C
Red Globe	Failure force	$a_{1}$	$a_{1}$	$a_{1}$	[5.92958,6.22342]	[5.94291,6.28771]	[5.84761,6.37055]
		$a_{2}$	$a_{2}$	$a_{2}$	[3.68198,3.96635]	[3.35101,4.03717]	[−632.9628,−588.61648]
		$x_{0}$	$x_{0}$	$x_{0}$	[0.01266,0.01374]	[0.00717,0.00719]	[142392547,142665452]
		dx	p	p	[0.00137,0.00197]	[1.33034,2.72470]	[0.35822,0.36674]
	Strain	$a_{1}$	$a_{1}$	$a_{1}$	[48.3140,55.3940]	[48.3192,52.6310]	[48.5088,51.56694]
		$a_{2}$	$a_{2}$	$a_{2}$	[10.6211,13.1275]	[8.78232,13.04670]	[11.87571,21.24001]
		$x_{0}$	$x_{0}$	$x_{0}$	[0.01481,0.01489]	[0.00968,0.01140]	[0.00775,0.00895]
		dx	p	p	[0.00150,0.00478]	[2.18486,3.67702]	[1.53576,3.04336]
	Penetration work	$y_{0}$	$a_{1}$	$a_{1}$	[12.6539,14.2127]	[19.8631,21.3265]	[20.6693,19.8532]
		$a_{1}$	$a_{2}$	$a_{2}$	[6.96141,9.43195]	[12.7285,14.9436]	[13.8749,15.69432]
		$t_{1}$	$x_{0}$	$x_{0}$	[0.01048,0.01064]	[0.00811,0.00825]	[0.00610,0.00686]
			p	p		[2.18028,2.56562]	[2.14975,2.32417]
Wink	Failure force	$y_{0}$	$a_{1}$	$a_{1}$	[1.15318,2.05972]	[4.24423,4.39719]	[688.75096,950.77796]
		$a_{1}$	$a_{2}$	$a_{2}$	[1.16327,1.29247]	[1.45481,2.96617]	[2.10452,3.38422]
		$t_{1}$	$x_{0}$	$x_{0}$	[0.00419,0.00433]	[0.00525,0.02017]	[−0.07274,−0.07204]
		$a_{2}$	p	dx	[1.46587,1.76237]	[1.30517,1.48073]	[0.01109,0.01185]
		$t_{2}$			[0.06545,0.06553]		
	Strain	$y_{0}$	$a_{1}$	$a_{1}$	[−4.7376,−3.3260]	[14.7637,16.0992]	[275.497,329.421]
		$a_{1}$	$a_{2}$	$a_{2}$	[18.4988,21.36154]	[2.81711,4.22225]	[2.36777,4.01813]
		$t_{1}$	$x_{0}$	$x_{0}$	[0.05806,0.05820]	[0.01140,0.01162]	[−0.05666,−0.05546]
			p	dx		[2.16533,3.01813]	[0.01749,0.01753]
	Penetration work	$y_{0}$	$a_{1}$	$a_{1}$	[−2.1603,−1.8374]	[11.76194,16.57234]	[12.5032,12.8534]
		$a_{1}$	$a_{2}$	$a_{2}$	[11.4343,17.2834]	[7.51074,8.66560]	[7.44383,8.42603]
		$t_{1}$	$x_{0}$	$x_{0}$	[0.15640,0.15706]	[0.00744,0.00750]	[0.00816,0.00820]
			dx	p		[0.00552,0.00556]	[1.67867,1.76279]

## Data Availability

Data is contained within the article.

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
