# Peer review of "Modeling Mathematical Relationship with Weight Loss and Texture on Table Grapes of ‘Red Globe’ and ‘Wink’ during Cold and Ambient Temperature Storage"

_foods, 2023, doi:10.3390/foods12132443_

Round 1
Reviewer 1 Report (Previous Reviewer 1)
The manuscript deals with modeling the mathematical relationship between weight loss and texture on two table grapes cultivars’ during cold and ambient temperature storage.
The English language must be revised.
Please read the manuscript carefully and correct all typos.
Please use “ºC” not “ºc” for temperature units.
Introduction
The topics must be better linked.
Materials and methods
“Approximately 3 kg of grapes were packed per bag using polyethylene (PE) bags (Anhui Tongcheng Xiangpeng Packaging Co., LTD) (thickness, 0.018 mm, 32 cm×54 cm) and stored at 0oC, 3 oC, and 13 oC, (RH 85 % ~90 %) for use.”??Why was 13 ºC selected??
Results and discussion
“During the 13 ºC temperature storage, the weight loss of 'Red Globe' was fairly higher than 'Wink' after the first 3 days, however, the weight loss of 'Red Globe' was significantly lower than 'Wink' during the middle and late storage stage (Fig. 1 A)”??Figure 1A needed??Please present only one figure with the experimental data and the fitting (as presented in Figure 1B, and identify each marker).
“The equations were showed in Table 1 and R2 correlation coefficients were all greater than 0.90, which means the fitting is significant.”??correlation coefficients?? Or coefficients of determination??Moreover, please add the values of adjusted-R2.
“So, the failure force of 'Red Globe' maintained a higher level than that of 'Wink' during the storage (Fig. 3A).”??Or Figure 2A??Moreover, please present only one figure with the experimental data and the fitting (as presented in Figure B, and identify each marker).
“78.59 %, 63.94 %, 50.25%; 38.56 %, 33.59 %, 28.72 % compared to 49.04 %, 35.01 %, 32.33 %; 79.34 %; 65.03 %, 56.06 %; 37.69 %, 34.68 %, 30.99 % in 'Wink' (Fig.4 A, C, E).”??Please present only one figure with the experimental data and the fitting (as presented in Figure B, and identify each marker).
Around 31 references have more than 5 years. Please update your list of references.
Please format the scientific names in italic.
The English language must be revised.
Author Response
Thank you for your comments concerning our manuscript entitled "Modeling mathematical relationship with weight loss and texture on two table grapes cultivars during cold and ambient temperature storage". These comments are very valuable in revising and improving our manuscript. We have carefully studied the comments and have made the necessary revisions as required. In addition, we corrected some errors, and the revised ports are marked in red in the manuscript. The responses to the reviewer’1 comments are listed as below:
Point 1:
The English language must be revised.
Please read the manuscript carefully and correct all typos.
Please use “ºC” not “ºc” for temperature units.
Response 1: Thank you for your comments. The English version has been proofread by a native speaker and we have corrected all typos and changed the “ºc” with “ºC”.
Point 2:
Introduction
The topics must be better linked.
Response 2: Thank you for your comments. We have revised the introduction to link the topics better.
Point 3:
Materials and methods
“Approximately 3 kg of grapes were packed per bag using polyethylene (PE) bags (Anhui Tongcheng Xiangpeng Packaging Co., LTD) (thickness, 0.018 mm, 32 cm×54 cm) and stored at 0 oC, 3 oC, and 13 oC, (RH 85 % ~90 %) for use.”??Why was 13 ºC selected??
Response 3: Thank you for your comments. The local ‘Red global’ and ‘Wink’ are harvested in mid-October and the room temperature is around 13 oC, therefore, we chose the 13 ºC for the study.
Point 4:
Results and discussion
“During the 13 ºC temperature storage, the weight loss of 'Red Globe' was fairly higher than 'Wink' after the first 3 days, however, the weight loss of 'Red Globe' was significantly lower than 'Wink' during the middle and late storage stage (Fig. 1 A)”??Figure 1A needed??Please present only one figure with the experimental data and the fitting (as presented in Figure 1B, and identify each marker).
Response 4: Thank you for your comments. During storage at 13 ºC, the weight loss of 'Red Globe' was slightly higher than that of 'Wink' in the first 3 days, but, the weight loss of 'Red Globe' was significantly lower than that of 'Wink' in the middle and late stages of storage, which may be caused by the storge environment and individual difference, and the two values have no significant difference. We have tried many times to show the variation and adaptation in one figure, but, if the parts were merged, the figure would be a bit of a mess, so we made two figures.
Point 5:
“The equations were showed in Table 1 and R2 correlation coefficients were all greater than 0.90, which means the fitting is significant.”??correlation coefficients?? Or coefficients of determination??Moreover, please add the values of adjusted-R2.
“So, the failure force of 'Red Globe' maintained a higher level than that of 'Wink' during the storage (Fig. 3A).”??Or Figure 2A??Moreover, please present only one figure with the experimental data and the fitting (as presented in Figure B, and identify each marker).
“78.59 %, 63.94 %, 50.25%; 38.56 %, 33.59 %, 28.72 % compared to 49.04 %, 35.01 %, 32.33 %; 79.34 %; 65.03 %, 56.06 %; 37.69 %, 34.68 %, 30.99 % in 'Wink' (Fig.4 A, C, E).”??Please present only one figure with the experimental data and the fitting (as presented in Figure B, and identify each marker).
Response 5: Thank you for your comments. R2 is the coefficient of determination, which measures how well a regression model fits the data. Its value is between 0 and 1, and the closer it is to 1, the better the model fits the data. The coefficients of determination were all greater than 0.90, so the fit is significant.
We have included the values of fitted-R2 in tables. We have tried many times to present the variation and fitting in one figure, but if the parts were merged, the figure would be a bit of a mess, so we made two figures.
The text “So, the failure force of 'Red Globe' maintained a higher level than that of 'Wink' during the storage (Fig. 3A).” has been changed to “Thus, the failure force of 'Red Globe' remained at a higher level than that of 'Wink' during the storage (Fig. 2A).”
“The loss of failure force, strain, penetration work in 'Red Globe' was 38.75 %, 34.50 %, 30.49 %; 78.59 %, 63.94 %, 50.25%; 38.56 %, 33.59 %, 28.72 % compared to 49.04 %, 35.01 %, 32.33 %; 79.34 %; 65.03 %, 56.06 %; 37.69 %, 34.68 %, 30.99 % in 'Wink'”, the part is used to demonstrate that 'Red Globe' maintained a higher level than that of 'Wink'.
Point 6:
Around 31 references have more than 5 years. Please update your list of references. Please format the scientific names in italic.
Response 6: Thank you for your comments. We have updated the reference and there are 30 references from 2019 and the scientific names are in italics.
We have tried our best to improve the manuscript and have made some changes in the manuscript. These changes do not affect the content and framework of the manuscript. We hope that our revision will meet the requirements.
Thank you for your kind considerations. We look forward to hearing from you soon.
With best regards,
Prof. Yang Bi
College of Food Science and Engineering,
Gansu Agricultural University,
Lanzhou 730030,
China.
E-mail: biyang@gsau.edu.cn
Tel: +86-931-7631113

Reviewer 2 Report (Previous Reviewer 2)
The paper was improved.
Author Response
Thanks very much for your comments.
Reviewer 3 Report (Previous Reviewer 3)
The manuscript has been revised comprehensively.
Author Response
Thanks very much for your comments.
Round 2
Reviewer 1 Report (Previous Reviewer 1)
The manuscript was improved.
Minor corrections are needed.
Author Response
Thank you for your comment concerning our manuscript entitled "Modeling mathematical relationship with weight loss and texture on table grapes of ‘Red Globe’ and ‘Wink’ during cold and ambient temperature storage". These comment are very valuable in revising and improving our manuscript. We have carefully studied the comment and have made the necessary revisions as required. In addition, we corrected some errors, and the revised ports are marked in ‘Track changes’ in the manuscript. The response to the reviewer’1 comment is listed as below:
Point 1:
Minor corrections are needed.
Response 1: Thank you for your comments. We corrected the manuscript carefully.
We have tried our best to improve the manuscript and have made corrections in the manuscript. These changes do not affect the content and framework of the manuscript. We hope that our revision will meet the requirements.
Thank you for your kind considerations. We look forward to hearing from you soon.
With best regards,
Prof. Yang Bi
College of Food Science and Engineering,
Gansu Agricultural University,
Lanzhou 730030,
China.
E-mail: biyang@gsau.edu.cn
Tel: +86-931-7631113

This manuscript is a resubmission of an earlier submission. The following is a list of the peer review reports and author responses from that submission.
Round 1
Reviewer 1 Report
The manuscript deals with modeling the mathematical relationship between weight loss and texture on two cultivars’ table grapes during storage.
The English language must be deeply revised.
Please read the manuscript carefully and correct all typos.
Please align all equations.
Please separate values from units, e.g. “2 mm” not “2mm”.
Please check the figures’ format, i.e. x-axis is confusing.
Introduction
This section must be improved. The topics must be better linked.
The introduction should briefly place the study in a broad context and highlight why it is important.
Materials and methods
Only one temperature tested??
Fruit characterization?
Results and discussion
This section has lack of depth and must be revised.
What is the starting point if the fruit was not characterized in terms of maturation??
The English language must be deeply revised.
Author Response
Point 1: The English language must be deeply revised.
Response 1: Thank you for your comments. The language of the text has been checked and improved by Mr. William Oyom, whose native language is English.
Point 2: Please read the manuscript carefully and correct all typos.
Response 2: We have read the the manuscript and corrected all the typos carefully.
Point 3: Please align all equations.
Response 3: We have aligned all the equations as suggested.
Point 4: Please separate values from units, e.g. “2 mm” not “2mm”.
Response 4: We have separated all the values from the units.
Point 5: Please check the figures’ format, i.e. x-axis is confusing.
Response 5: Thanks. We have checked all the figures and unified the x-axis’ ticks from 1 to 13 with the same interval of 2 at the storage time.
Point 6: Introduction
This section must be improved. The topics must be better linked.
The introduction should briefly place the study in a broad context and highlight why it is important.
Response 6: Thanks for your comments. Accordingly , we revised the introduction from human health and enviromental protection and highlighted the improtance of the topics.
Point 7: Materials and methods
Only one temperature tested??
Fruit characterization?
Response 7: Thank you spotting this. The ambient temperature storage of the fruits was chosen in line with the investigation period of our study. Since ambient conditions accelerate weight loss and physiological changes, this condition was more useful than the refrigeration storage. We have added the fruit characterization in the materials and methods section.
Point 8: Results and discussion
This section has lack of depth and must be revised.
What is the starting point if the fruit was not characterized in terms of maturation??
Response 8: We have revised the results and discussion according to the suggestion. The matuation (obrix: Red Globe, 17 % - 18 %, Wink, 15 % - 16%; total acid: Red Globe, 6.206. g/ L- 6.296 g / L, Wink, 5.769 g / L – 6.522 g / L; sugar-acid ratio: Red Globe, 27 - 29, Wink, 23 – 26; firmness: Red Globe, 128.299 g.mm, Wink, 161.049 g.mm ) was characterized as the starting point of the grapes. As the main topic of the article is to figure out the mathematical relationship between weight loss and texture, so the maturation is not discussed deeply in the result.
Reviewer 2 Report
The paper was improved.
Author Response
Thanks a lot for your comments.
Reviewer 3 Report
I have closely read through the manuscript and am satisfied with the revised version. The paper was improved.
Author Response
Thanks a lot for your comments.